# Impact of Preoperative Malnutrition on Postoperative Quality of Life in Older Adults Undergoing Surgery for Degenerative Cervical Myelopathy: A Retrospective Cohort Study

**DOI:** 10.3390/nu17182912

**Published:** 2025-09-09

**Authors:** Yuki Taniguchi, Hideki Nakamoto, So Kato, Hiroyuki Nakarai, Kosei Nagata, Kenichi Kono, Yuhei Saito, Reo Inoue, Hiroshi Okawa, Sakae Tanaka, Yasushi Oshima, Kazuhiko Fukatsu

**Affiliations:** 1Surgical Center, The University of Tokyo Hospital, Tokyo 113-8655, Japan; kkouno_tki@yahoo.co.jp (K.K.); fukatsu-1su@h.u-tokyo.ac.jp (K.F.); 2Department of Orthopaedic Surgery, The University of Tokyo Hospital, Tokyo 113-8655, Japan; hidekinski.indifferent@gmail.com (H.N.); skatou-tky@umin.net (S.K.); h.nakarai.ort@gmail.com (H.N.); koseinagata1984@gmail.com (K.N.); tanakas-ort@h.u-tokyo.ac.jp (S.T.); yoo-tky@umin.ac.jp (Y.O.)

**Keywords:** malnutrition, health-related quality of life, geriatric, degenerative cervical myelopathy

## Abstract

**Background/Objectives:** Malnutrition, which is closely associated with frailty and sarcopenia, is common in older adults and is linked to adverse perioperative complications in musculoskeletal surgery. However, its influence on postoperative health-related quality of life (HRQOL) remains unclear. This study aimed to investigate the impact of preoperative malnutrition on HRQOL one year after surgery in elderly patients with degenerative cervical myelopathy (DCM). **Methods:** We retrospectively analyzed 188 patients aged ≥ 65 years who underwent elective surgery for DCM between 2017 and 2024. Preoperative nutritional status was assessed using the Geriatric Nutritional Risk Index (GNRI), with GNRI ≤ 98 indicating malnutrition risk. Patient-reported outcome measures were assessed using the EuroQol Five-Dimension Questionnaire (EQ-5D) both preoperatively and at one year postoperatively. The minimum clinically important difference (MCID) threshold was applied to evaluate significant changes. Multivariate logistic regression was used to identify independent risk factors for postoperative deterioration in EQ-5D score. **Results:** Of the 188 patients, 35 were classified as having malnutrition risk. While preoperative EQ-5D scores were comparable between the two groups, the postoperative EQ-5D score was significantly lower in the malnutrition risk group than in the no-risk group (0.58 vs. 0.67, *p* = 0.003). Deterioration in EQ-5D scores exceeding the MCID threshold occurred more frequently in the malnutrition risk group (37.1% vs. 21.2%, *p* = 0.049). Furthermore, multivariate analysis identified preoperative GNRI ≤ 98 as an independent risk factor for deterioration in EQ-5D score exceeding the MCID threshold (OR 2.40, 95% CI 1.03–5.52). **Conclusions:** Preoperative malnutritional status was significantly associated with impaired postoperative HRQOL in elderly patients with DCM. These findings highlight the need for preoperative nutritional assessment and optimization in this vulnerable population.

## 1. Introduction

Population aging is progressing globally, drawing increasing attention to the concept of health-adjusted life expectancy [1,2]. Consequently, more elderly individuals are becoming candidates for surgical treatment. In this context, the ability to maintain activities of daily living (ADLs) and preserve quality of life (QOL) after surgery has become an increasingly critical concern. Importantly, it is well established that older adults have an increased risk of malnutrition, which is closely associated with sarcopenia and frailty [3,4,5]. Therefore, it is imperative to elucidate the impact of malnutrition, particularly in the context of musculoskeletal treatment, in the elderly population. In recent years, optimizing diet quality to prevent malnutrition has been shown to reduce the risk of decline in physical performance in older adults [6]. In the perioperative management of musculoskeletal surgery, the importance of immunonutrition has attracted increasing attention, and perioperative optimization of nutritional status has been reported to reduce postoperative complications and shorten the length of hospital stay [7,8]. Thus, the need for early screening and multimodal, multidisciplinary interventions for perioperative malnutrition has been increasingly emphasized [9]. In spinal surgery, preoperative malnutrition has been repeatedly reported as a risk factor for higher perioperative complication rates [10,11,12,13]. However, its impact on postoperative health-related quality of life (HRQOL) remains largely underexplored. In particular, in patients with degenerative cervical myelopathy (DCM), surgical intervention is often unavoidable, even in those with advanced age or poor general health, as conservative management may result in severe neurological deterioration, including quadriparesis. Therefore, it is crucial to investigate the extent to which preoperative malnutrition affects postoperative recovery of QOL. Based on the hypothesis that preoperative malnutrition has a significant impact on postoperative recovery of ADLs in older adults with DCM, we conducted this study to examine the influence of preoperative nutritional status on HRQOL 1 year after surgery in patients aged ≥65 years.

## 2. Materials and Methods

### 2.1. Data Source and Patients

We conducted a retrospective analysis of our prospective cohort, enrolling consecutive patients aged ≥65 years who underwent elective surgery for DCM in our hospital between 1 April 2017 and 31 January 2024. We excluded patients who underwent revision surgery, those with rheumatoid arthritis, those on hemodialysis, those with atlantoaxial subluxation, and individuals from whom informed consent could not be obtained. This study was approved by the Research Ethics Committee of the University of Tokyo [approval numbers 10335-(9) and 2674-(9)], and written informed consent was obtained from all participants.

### 2.2. Data Collection

The collected data included patient characteristics, surgery-related information, and patient-reported outcome measures (PROMs). Patient characteristics included age, sex, body mass index (BMI), diagnosis, preoperative American Society of Anesthesiologists Physical Status (ASA-PS) score, smoking status, and diagnosis of diabetes mellitus. Diagnosis of the disease was classified into cervical spondylotic myelopathy (CSM), ossification of the posterior longitudinal ligament (OPLL), and cervical disc herniation. For univariate and multivariate analyses, patients’ ages were categorized into three groups: 65–74, 75–84, and ≥85.

Surgery-related data included operation time, estimated blood loss, surgical procedure, and use of microendoscopy. Surgical procedures included posterior decompression, such as laminectomy and laminoplasty, posterior decompression with fusion, anterior decompression with fusion, and combined anterior and posterior decompression with fusion. The occurrence of incidental intraoperative durotomy was also recorded.

For PROMs, we used the EuroQol Five-Dimension Questionnaire (EQ-5D) administered preoperatively and one year postoperatively as the primary outcome [14]. For the subsequent analysis, EQ-5D health states were converted into a single summary number [15]. Based on a previous report, the threshold for the minimum clinically important difference (MCID) for EQ-5D was set at 0.0485 [16]. Additionally, postoperative patient satisfaction was evaluated one year after surgery using a seven-point Likert scale (very satisfied, satisfied, slightly satisfied, neither satisfied nor dissatisfied, slightly dissatisfied, dissatisfied, and very dissatisfied) [17,18]. Patients who stated that they were very satisfied, satisfied, or slightly satisfied were classified as satisfied, whereas those who reported any other response were categorized as dissatisfied. Missing entries in the preoperative or one-year postoperative EQ-5D assessments were treated as missing values. In the present study, preoperative EQ-5D assessment was missing in three cases, and postoperative EQ-5D assessment at one year was missing in five cases. In accordance with the default settings of the statistical software, both univariate and multivariate analyses were performed using complete-case analysis (listwise deletion), and cases with missing values were excluded.

### 2.3. Assessment of Preoperative Nutritional Status

Preoperative nutritional status was evaluated using the Geriatric Nutritional Risk Index (GNRI), which was calculated from the patient’s preoperative serum albumin level and BMI [19]. Patients with a preoperative GNRI ≤ 98 were classified as having malnutrition risk, as previously described [20,21,22].

### 2.4. Statistical Analysis

Continuous variables were compared using *t*-tests. The chi-squared test was used to analyze categorical data. Multivariate logistic regression analysis was performed to identify risk factors associated with a decline in EQ-5D health status one year after surgery that exceeded the MCID. Explanatory variables for the multivariate analysis were selected based on those with low *p*-values in the univariate analysis, in addition to basic demographic factors, including sex and age. The threshold for significance was set at *p* < 0.05. All statistical analyses were performed using JMP Student Edition 18 (SAS Institute Inc., Cary, NC, USA).

## 3. Results

### 3.1. Patient Characteristics

We identified 188 eligible patients in our cohort, comprising 110 males and 78 females. Table 1 provides detailed information regarding the study population. The mean age of patients at the time of surgery was 76.1 years, and their mean BMI was 24.1 kg/m^2^ (Table 1). The mean preoperative GNRI was 106.1, indicating an overall good preoperative nutritional status in the study population (Table 1). Diagnoses included CSM in 128 (68.1%), OPLL in 55 (29.3%), and cervical disc herniation in 5 (2.7%) patients (Table 1). Diabetes was present in 62 patients (33.0%), and most (81.4%) were preoperatively categorized as ASA-PS 2 (Table 1). Regarding surgical procedures, the majority of patients (88.8%) underwent posterior decompression, and 28 patients (14.9%) underwent microendoscopic surgery (Table 1). Incidental durotomy was identified in six patients (3.2%) (Table 1). The mean EQ-5D score for the whole study population significantly improved from 0.57 to 0.66 (*p* < 0.0001) (Table 1).

### 3.2. Comparison Between the Malnutrition Risk and No-Risk Groups

The malnutrition risk group (those with preoperative GNRI ≤ 98) included 35 patients, whereas the remaining 153 patients (with preoperative GNRI > 98) were categorized into the no-risk group (Table 2). Regarding patient characteristics, patients in the malnutrition risk group had a significantly lower BMI, as expected (Table 2). Although there was a significant difference in the distribution of ASA-PS between the two groups, no significant differences were observed in other baseline characteristics (Table 2). Regarding PROMs, there was no significant difference between the groups in terms of patient satisfaction at the one-year assessment (Table 2). In contrast, although the preoperative EQ-5D values were comparable between the groups, the score at one year postoperatively was significantly lower in the malnutrition risk group than in the no-risk group (0.58 vs. 0.67, *p* = 0.003) (Table 2). In the malnutrition risk group, there was minimal improvement in EQ-5D values compared with the preoperative level (0.57 preoperatively vs. 0.58 one year postoperatively) (Table 2).

### 3.3. Proportions of Patients with Changes Exceeding the MCID Thresholds for EQ-5D Score in the Malnutrition Risk and No-Risk Groups

Subsequently, we analyzed the proportion of patients whose EQ-5D score changes exceeded the MCID thresholds, either for improvement or deterioration. Although there was no significant difference in the proportion of patients who achieved an improvement exceeding the MCID, the proportion of patients who experienced deterioration beyond the MCID was significantly higher in the malnutrition risk group than in the no-risk group (37.1% vs. 21.2%, *p* = 0.049) (Table 3).

### 3.4. Risk Factors for Deterioration in EQ-5D Score Exceeding the MCID Threshold

To determine the actual impact of preoperative nutritional status on postoperative EQ-5D-deterioration, we conducted univariate and multivariate analyses. Univariate analysis did not reveal statistically significant associations, but incidental durotomy and preoperative GNRI ≤ 98 tended to be associated with a deterioration in EQ-5D exceeding the MCID (Table 4). Multivariate logistic regression analysis identified preoperative GNRI ≤ 98 as an independent risk factor for deterioration in EQ-5D values exceeding the MCID threshold at one year postoperatively (odds ratio, 2.40; 95% confidence interval, 1.03–5.52) (Table 4). Subsequently, as a sensitivity analysis, we performed multivariate logistic regression analyses, excluding each covariate one at a time. The significance of preoperative GNRI ≤ 98 became marginal only when age was removed from the model; in all other models, it remained statistically significant, thereby supporting the robustness of the present findings (Appendix A).

## 4. Discussion

This study demonstrates that preoperative malnutrition risk has a clear impact on postoperative HRQOL in elderly patients with DCM. Patients with preoperative malnutrition risk, defined as GNRI ≤ 98, had significantly lower EQ-5D values one year after surgery than those without malnutrition risk. Furthermore, preoperative GNRI ≤ 98 was identified as an independent risk factor for deterioration in EQ-5D score exceeding the MCID threshold at one year postoperatively.

Previous studies have demonstrated a relationship between preoperative nutritional status and perioperative complications following cervical spine surgery. Lee et al. found that a baseline serum albumin <3.5 g/dL might serve as a predictor of several complications following posterior cervical fusion [13]. Kurosu et al. identified the Prognostic Nutritional Index (PNI) as an indicator of perioperative medical complications in cervical posterior surgery [23]. Alam et al. revealed that malnutrition, determined using GNRI, was an independent risk factor for 30-day complications, readmissions, prolonged hospital stay, and non-home discharge following anterior cervical discectomy and fusion in elderly patients [24]. Furthermore, severe malnutrition, as defined using GNRI, was an independent risk factor for postoperative mortality in elderly patients undergoing posterior cervical decompression/fusion [25]. However, no reports have demonstrated an association between preoperative nutritional status and postoperative HRQOL following such surgeries. In recent years, a small number of studies in other musculoskeletal domains have reported that preoperative malnutrition adversely affects postoperative HRQOL. Kokubu et al. demonstrated that preoperative malnutrition, defined as GNRI ≤ 98, was a significant risk factor for long-term deterioration of PROMs, as assessed by the Knee Society Score following total knee arthroplasty [26]. Chen et al., in a prospective cohort study of elderly patients with hip fracture, reported that patients in the higher PNI group had significantly higher EQ-5D scores at the 30-day follow-up and a significantly greater likelihood of achieving unrestricted mobility at the 120-day follow-up [27]. Both studies concluded that preoperative malnutrition negatively influenced postoperative HRQOL, which is consistent with the findings of the present study. Importantly, similar to our investigation, these studies focused on elderly patients at the time of follow-up, suggesting that nutritional status may play a particularly crucial role in the recovery of musculoskeletal function in older adults.

Malnutrition has been closely linked to sarcopenia and frailty, and recent reports have highlighted the relationship between sarcopenia/frailty and postoperative HRQOL following cervical spine surgery [28,29,30]. Concerning sarcopenia, some reports have demonstrated an association between preoperative paraspinal sarcopenia and functional outcomes following cervical spine surgery [31,32]. For frailty, the five-item Modified Frailty Index was identified as a significant predictor of long-term postoperative outcomes in patients undergoing cervical posterior laminoplasty for OPLL [33]. Another study found that preoperative frailty significantly reduced the chance of achieving the MCID for postoperative functional impairment following surgery for DCM [34]. Given the close relationship between malnutrition and frailty/sarcopenia and the impact of frailty/sarcopenia on postoperative outcomes in DCM patients, the findings of the present study appear to be reasonable. It is plausible to assume that in some elderly patients with preoperative malnutrition risk, the impact of age-related sarcopenia may outweigh the benefits of neurological recovery, ultimately leading to a decline in ADLs. However, because data pertaining to body-composition measures were unavailable in this study, the extent to which sarcopenia contributed to the outcomes in the malnutrition group remains uncertain. Nevertheless, recent methodological advances have enabled the use of anthropometric data in combination with artificial intelligence to predict muscle mass loss and sarcopenia [35]. As such approaches are expected to facilitate more straightforward preoperative assessment of sarcopenia in the near future, further research is warranted to clarify how sarcopenia and malnutrition interact with each other and collectively influence postoperative outcomes.

Another possible factor contributing to the poor postoperative outcomes observed in patients with preoperative malnutrition risk in this study may be related to cervical alignment. A previous study demonstrated that preoperative malnutrition is a risk factor for postoperative cervical kyphosis development in elderly patients undergoing laminoplasty for CSM [36]. Postoperative cervical malalignment has also been reported to be associated with suboptimal surgical outcomes [37,38]. Since the majority of patients in our cohort underwent posterior decompression procedures, including laminoplasty, it is possible that deterioration of cervical alignment following surgery in patients at risk of malnutrition contributed to the poorer HRQOL outcomes observed. However, because radiographic parameters were not analyzed in this study, this hypothesis remains speculative. Future analyses incorporating radiographic findings may be necessary to further elucidate the impact of preoperative malnutrition on postoperative HRQOL.

There is an ongoing debate regarding the optimal indicator for assessing preoperative nutritional status. Various indices have been proposed in the literature, including GNRI, PNI, the Mini Nutritional Assessment–Short Form, the Controlling Nutritional Status (CONUT) score, the Global Leadership Initiative on Malnutrition criteria, and the Malnutrition Universal Screening Tool [19,39,40,41,42,43]. A previous study showed that GNRI demonstrated superior predictive ability for perioperative adverse medical events compared to PNI and CONUT scores in older adults undergoing spinal surgery [44]. Their findings motivated us to adopt GNRI as a nutritional status index in the present study. Nevertheless, further investigation is warranted to identify the nutritional index that best correlates with postoperative HRQOL in patients undergoing surgery for DCM.

Our findings support the incorporation of routine nutritional risk screening into the preoperative pathway for elderly patients undergoing surgery for DCM. Practical approaches include brief clinic-based screening and/or the use of validated indices such as the GNRI, followed by early referral to a registered dietitian for targeted counseling and initiation of oral nutritional supplementation when risk is identified [45]. An important unresolved question is whether preoperative nutritional optimization can improve the prognosis of nutritionally at-risk patients undergoing surgery for DCM. Addressing this question will require prospective, ideally multicenter, studies. A pragmatic randomized trial could allocate patients with GNRI ≤ 98 to usual care or to a standardized nutrition bundle that includes dietitian-led counseling and high-protein oral nutritional supplementation initiated at least 7 days preoperatively, when feasible.

This study has several limitations. First, its retrospective, single-center design is prone to information bias and residual confounding. Because exposures and covariates were abstracted from routine clinical records rather than collected under a prespecified protocol, misclassification is possible. Important confounders—such as the severity of frailty/sarcopenia, socioeconomic factors, intensity of perioperative rehabilitation, and detailed radiographic parameters—were not fully captured, introducing the potential for unmeasured or inadequately measured confounding that could bias estimates in either direction. Second, selection bias cannot be excluded. Eligibility for elective surgery depends on the medical fitness of the patient and clinician judgment; patients with more severe malnutrition or complex conditions may have been triaged to nonoperative care or deferred for optimization and thus not included, which could attenuate the observed associations. Conversely, preoperative optimization among those proceeding to surgery may have selectively improved nutritional indices. Third, the sample size was relatively modest in relation to the number of covariates and events, limiting statistical power for subgroup analyses and yielding wider confidence intervals. Small samples also increase model instability and reduce the ability to explore non-linear effects or interactions, thereby constraining the precision and generalizability of the estimates.

## 5. Conclusions

The present study is the first to demonstrate the negative impact of preoperative malnutrition status, defined as GNRI ≤ 98, on postoperative HRQOL in elderly patients with DCM. Patients with preoperative malnutrition risk had significantly lower EQ-5D values one year after surgery than those without malnutrition risk. Furthermore, preoperative malnutrition risk was identified as an independent risk factor for deterioration in EQ-5D score, exceeding the MCID threshold at one year postoperatively. These findings underscore the importance of preoperative nutritional assessment and management in geriatric patients undergoing cervical spine surgery.

## Figures and Tables

**Table 1 nutrients-17-02912-t001:** Patient characteristics, surgery-related data, and patient-reported outcome measures of the study population.

Number of Patients	188	
Sex		
Male	110	(58.5)
Female	78	(41.5)
Age at surgery, years	76.1	[6.0]
Body mass index, kg/m^2^	24.1	[3.8]
Preoperative GNRI	106.1	[8.7]
Diagnosis		
Cervical spondylotic myelopathy	128	(68.1)
Ossification of the posterior longitudinal ligament	55	(29.3)
Cervical disc herniation	5	(2.7)
ASA-PS		
1	11	(5.9)
2	153	(81.4)
3	23	(12.2)
4	1	(0.5)
Smoking	13	(6.9)
Diabetes	62	(33.0)
Operation time, min	144.4	[69.1]
Estimated blood loss, mL	121.8	[182.2]
Surgical procedure		
Posterior decompression	167	(88.8)
Posterior decompression with fusion	17	(9.0)
Anterior decompression with fusion	3	(1.6)
Anterior and posterior fusion with decompression	1	(0.5)
Microendoscopic surgery	28	(14.9)
Incidental durotomy	6	(3.2)
EQ-5D value		
Before surgery	0.57	[0.18]
One year after surgery	0.66	[0.17]

Values are presented as *n* (%) or mean [SD]. SD, standard deviation; GNRI, Geriatric Nutritional Risk Index; ASA-PS, American Society of Anesthesiologists Physical Status; EQ-5D, EuroQol Five-Dimension Questionnaire.

**Table 2 nutrients-17-02912-t002:** Comparison between the malnutrition risk and no-risk groups.

	Malnutrition Risk Group (GNRI ≤ 98)	No-Risk Group (GNRI > 98)	*p*
Number of patients (%)	35 (18.6)	153 (81.4)	
Sex (Male:Female)	19:16	91:62	0.58
Age at surgery, years	77.3 [6.7]	75.8 [5.8]	0.18
Body mass index, kg/m^2^	20.3 [2.3]	24.9 [3.5]	**<0.001**
Preoperative GNRI	93.9 [3.9]	108.8 [6.9]	**<0.001**
Diagnosis			
(CSM:OPLL:CDH)	25:8:2	103:47:3	0.34
ASA-PS (1:2:3:4)	3:24:7:1	8:129:16:0	**0.048**
Smoking (yes:no)	3:32	10:143	0.71
Diabetes (yes:no)	23:12	103:50	0.84
Operation time, min	136.9 [42.9]	146.1 [72.0]	0.48
Estimated blood loss, mL	97.0 [102.2]	127.5 [195.8]	0.37
Surgical procedure			
(PD:PDF:ADF:APDF)	30:4:1:0	137:13:2:1	0.81
Microendoscopic surgery (yes:no)	3:32	25:128	0.30
Incidental durotomy (yes:no)	1:34	5:148	1.00
Patients’ satisfaction			
(satisfied:dissatisfied)	21:14	104:49	0.43
EQ-5D value			
Before surgery	0.57 [0.20]	0.57 [0.17]	0.99
One year after surgery	0.58 [0.16]	0.67 [0.16]	**0.003**

Values are presented as *n* (%) or mean [SD]. GNRI, Geriatric Nutritional Risk Index; SD, standard deviation; CSM, cervical spondylotic myelopathy; OPLL, ossification of the longitudinal ligament; CDH, cervical disc herniation; ASA-PS, American Society of Anesthesiologists Physical Status; PD, posterior decompression; PDF, posterior decompression with fusion; ADF, anterior decompression with fusion; APDF, anterior and posterior decompression with fusion; EQ-5D, EuroQol Five-Dimension Questionnaire.

**Table 3 nutrients-17-02912-t003:** The number of patients in the malnutrition risk and no-risk groups whose one-year changes in EuroQol Five-Dimension Questionnaire (EQ-5D) score reached the minimum clinically important difference.

	Malnutrition Risk Group (GNRI ≤ 98)	No-Risk Group (GNRI > 98)	*p*
One-year change in EQ-5D value (%)			
∆ ≥ 0.0485	16/35 (45.7)	85/146 (58.2)	0.19
∆ ≤ −0.0485	13/35 (37.1)	31/146 (21.2)	**0.049**

GNRI, Geriatric Nutritional Risk Index; ∆, changes in each patient-reported outcome measure from preoperative baseline to one year postoperatively.

**Table 4 nutrients-17-02912-t004:** Univariate and multivariate analyses for deterioration in EQ-5D score exceeding the MCID of −0.0485 at one year postoperatively.

	Univariate Analysis		Multivariate Analysis	
	Odds Ratio [CI]	*p*	Odds Ratio [CI]	*p*
Age, years		0.94		0.71
85+	0.88 [0.18–3.23]		0.68 [0.13–2.68]	
75–84	1.09 [0.54–2.26]		1.19 [0.57–2.50]	
65–74	Reference		Reference	
Sex		0.87		0.79
Male	1.06 [0.53–2.14]		1.10 [0.55–2.27]	
Female	Reference		Reference	
Preoperative GNRI		0.06		**0.04**
≤98	2.19 [0.98–4.81]		2.40 [1.03–5.52]	
>98	Reference		Reference	
Diabetes		0.97		
Yes	0.99 [0.47–2.02]			
No	Reference			
Smoking		0.24		
Yes	2.07 [0.60–6.56]			
No	Reference			
Diagnosis		0.32		
OPLL	1.01 [0.47–2.10]			
CDH	<0.001 [*–1.93]			
CSM	Reference			
ASA-PS		0.27		
3 or 4	4.80 [0.70–96.7]			
2	2.79 [0.50–52.3]			
1	Reference			
Surgical procedure		0.94		
Fusion with decompression	1.04 [0.32–2.89]			
Decompression alone	Reference			
Microendoscopic surgery		0.44		
Yes	0.67 [0.21–1.77]			
No	Reference			
Incidental durotomy		0.06		0.07
Yes	<0.001 [*–1.15]		<0.001 [*–1.16]	
No	Reference		Reference	

*, not estimable; EQ-5D, EuroQol Five-Dimension Questionnaire; CI, confidence interval; OPLL, ossification of the longitudinal ligament; CDH, cervical disc herniation; CSM, cervical spondylotic myelopathy; ASA-PS, American Society of Anesthesiologists Physical Status; GNRI, Geriatric Nutritional Risk Index.

## Data Availability

The data presented in this study are available from the corresponding author upon request due to restrictions related to the protection of personal information.

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
