# Peer review of "Impact of Preoperative Malnutrition on Postoperative Quality of Life in Older Adults Undergoing Surgery for Degenerative Cervical Myelopathy: A Retrospective Cohort Study"

_nutrients, 2025, doi:10.3390/nu17182912_

Round 1
Reviewer 1 Report
Comments and Suggestions for Authors
Dear Authors,
This study examined the influence of preoperative nutritional status on HRQOL one year after surgery in patients aged ≥65 years.
Here my comments:
-
Title: Please specify the design of the study.
-
Main text: Use a more formal academic language, for example by avoiding the use of “we” and “our.”
-
Discussion: Consider adding that nowadays artificial intelligence can facilitate the screening of muscle mass loss (DOI: 10.1016/j.heliyon.2023.e16323).”
The English could be improved to more clearly express the research.
Reviewer 2 Report
Comments and Suggestions for Authors
This article discusses the importance of malnutrition in older patients undergoing surgery for degenerative cervical myelopathy. It is interesting because it raises a very important topic – the consequences of malnutrition on older patients referred for spine surgery.
-In the introduction, the authors mention the importance of malnutrition too briefly, please expand on this aspect.
- The discussion part highlighted the association of malnutrition with sarcopenia and frailty, as well as the risk of cervical alignment. This study lacks BMD data and assessment of cervical alignment, making it difficult to assume that these changes concern only the malnutrition group.
Reviewer 3 Report
Comments and Suggestions for Authors
The article is very interesting, thank you . There are some areas of improvement that will contribute to its scientific value:
First, a visual summary of the study design and main findings would enhance accessibility and impact, helping readers quickly grasp the clinical message. While the discussion is solid, it could be strengthened by providing a deeper comparison with existing studies on malnutrition and surgical outcomes in other specialties (orthopedics, oncology, gastrointestinal surgery), further elaborating on how the present results align with or diverge from earlier research in spinal surgery populations. Some thing else are the clinical implications: The manuscript would benefit from a more explicit discussion of how nutritional screening and potential interventions (e.g., supplementation, dietary counseling) could be integrated into preoperative care pathways.
Also, the authors could suggest avenues for prospective studies or interventional trials to evaluate whether nutritional optimization can improve postoperative HRQOL in elderly patients.
Methodological clarifications are also needed. More detail on the handling of missing data for patient-reported outcomes would improve transparency. Additionally, sensitivity analyses or subgroup explorations could further support the robustness of the findings.
Limitations: Although limitations are mentioned, a more detailed reflection on the retrospective design, potential selection bias (e.g., exclusion of patients with poor nutritional status not eligible for surgery), and the relatively small sample size would strengthen the credibility of the conclusions.
Overall, congrats for the good work.
Round 2
Reviewer 1 Report
Comments and Suggestions for Authors
Dear Authors,
I appreciated your efforts in improving manuscript. In my opinion, it could be considered for publication.